# Cryo-EM reveals new species-specific proteins and symmetry elements in the *Legionella pneumophila* Dot/Icm T4SS

**Michael J Sheedlo[1,2†], Clarissa L Durie[3†], Jeong Min Chung[3,4], Louise Chang[3], Jacquelyn Roberts[3], Michele Swanson[5], Dana Borden Lacy[2,6], Melanie D Ohi[3,7*]**

[1]Department of Pharmacology, University of Minnesota, Minneapolis, United States; [2]Department of Pathology, Microbiology, and Immunology, Vanderbilt University Medical Center, Nashville, United States; [3]Life Sciences Institute, University of Michigan, Ann Arbor, United States; [4]Department of Biotechnology, The Catholic University of Korea, Gyeonggi, Republic of Korea; [5]Department of Microbiology and Immunology, University Of Michigan, Ann Arbor, United States; [6]The Veterans Affairs Tennessee Valley Healthcare System, Nasvhille, United States; [7]Department of Cell and Developmental Biology, University of Michigan, Ann Arbor, United States

**\*For correspondence:**
mohi@umich.edu

[†]These authors contributed equally to this work

**Competing interest:** The authors declare that no competing interests exist.

**Abstract** *Legionella pneumophila* is an opportunistic pathogen that causes the potentially fatal pneumonia known as Legionnaires' disease. The pathology associated with infection depends on bacterial delivery of effector proteins into the host via the membrane spanning Dot/Icm type IV secretion system (T4SS). We have determined sub-3.0 Å resolution maps of the Dot/Icm T4SS core complex by single particle cryo-EM. The high-resolution structural analysis has allowed us to identify proteins encoded outside the Dot/Icm genetic locus that contribute to the core T4SS structure. We can also now define two distinct areas of symmetry mismatch, one that connects the C18 periplasmic ring (PR) and the C13 outer membrane cap (OMC) and one that connects the C13 OMC with a 16-fold symmetric dome. Unexpectedly, the connection between the PR and OMC is DotH, with five copies sandwiched between the OMC and PR to accommodate the symmetry mismatch. Finally, we observe multiple conformations in the reconstructions that indicate flexibility within the structure.

## Introduction

Type IV secretion systems (T4SS) are large molecular machines utilized by many bacteria and some archaea. Several pathogenic bacteria, such as *Legionella pneumophila*, *Helicobacter pylori*, *Bordetella pertussis*, *Brucella*, and *Bartonella*, use T4SSs to deliver bacterial molecules (either nucleic acids or proteins) into the cytoplasm of their host (*Christie et al., 2014*; *Grohmann et al., 2018*). The activity of these effector molecules contributes to a variety of human diseases including pneumonia, gastric cancer, whooping cough, and 'cat scratch fever' (*Christie et al., 2014*; *Grohmann et al., 2018*).

T4SSs in Gram-negative bacteria contain a minimum of 12 components (named VirB1-VirB11 and VirD4 in prototype systems), which are organized into a structure that spans both the inner and outer membranes. The architecture of T4SSs can be subdivided into at least four different features: the outer membrane core or cap (OMC), an inner membrane complex, a complement of cytosolic ATPases, and, in some species, an extracellular pilus (*Christie et al., 2014*; *Grohmann et al., 2018*; *Fronzes et al., 2009*; *Gordon et al., 2017*; *Waksman, 2019*; *Low et al., 2014*; *Gonzalez-Rivera et al., 2016*). Though several of these features are conserved among species, the exact architecture varies between systems. For example, recent structural studies of the *L. pneumophila* Dot/Icm and *H. pylori* Cag T4SSs revealed a periplasmic ring (PR) that had not been identified in 'minimized' systems

(*Ghosal et al., 2017*; *Ghosal et al., 2019*; *Chetrit et al., 2018*; *Park et al., 2020*; *Chang et al., 2018*; *Chung et al., 2019*; *Sheedlo et al., 2020*; *Durie et al., 2020*; *Hu et al., 2019*). A symmetry mismatch between the OMC and PR was also described for both systems, with a C13:C18 (OMC:PR) mismatch in the *L. pneumophila* Dot/Icm T4SS and a C14:C17 (OMC:PR) mismatch in the *H. pylori* Cag T4SS. Though the connections between the OMC and the PR could not be modeled for either system, it was discovered that the *H. pylori* VirB9 homolog (known as CagX) is present in both the OMC and PR, leading to questions regarding how the symmetry mismatch is accommodated in these systems. The PR is distinct among the structurally characterized T4SSs, though a similar symmetry mismatch phenomenon has been described in bacterial type II (T2SS), type III (T3SS), and type VI (T6SS) secretion systems, suggesting a utility to its conservation (*Ghosal et al., 2019*; *Chung et al., 2019*; *Sheedlo et al., 2020*; *Durie et al., 2020*; *Chernyatina and Low, 2019*; *Hu et al., 2018*; *Dix et al., 2018*).

The Dot/Icm complex of *L. pneumophila* is one of the largest known T4SSs. Genetic screens for mutants defective in intracellular replication identified 26 genes, named *dot* (defect in organelle trafficking) or *icm* (intracellular multiplication), required for T4SS function (*Segal et al., 1998*; *Segal and Shuman, 1999*; *Berger and Isberg, 1993*; *Vogel et al., 1998*). The Dot/Icm T4SS has as many as 300 protein substrates, a striking contrast to the *H. pylori* Cag and *B. pertussis* Ptl T4SSs, each of which transports only one virulence factor (*Schroeder, 2017*; *Fischer, 2011*; *Backert et al., 2017*; *Shrivastava and Miller, 2009*). The Dot/Icm 'core complex' (which spans the inner and outer membranes) was originally predicted to contain only five proteins: DotC, DotD, DotF, DotG, and DotH (*Kubori et al., 2014*; *Nagai and Kubori, 2011*). We recently described parts of the structures and positions of DotC, DotD, and DotH, as well as two additional proteins that associate with the OMC, DotK, and Lpg0657 (*Ghosal et al., 2019*; *Durie et al., 2020*; *Kubori et al., 2014*). However, several features in this map could not be unambiguously identified, including the PR, three chains within the OMC, and a low-resolution 'dome' positioned in the center of the OMC which we predict breaches the outer membrane (*Durie et al., 2020*). Because the dome could not be resolved, we were unable to model the Dot/Icm T4SS pore that facilitates the transfer of cargo across the outer membrane. Structural and mass spectrometry analysis of T4SS particles purified from a deletion mutant strain revealed that two of the unidentified chains within the OMC were not DotG or DotF, because the structural organization of these unassigned regions in the mutant T4SS were unchanged compared to the WT T4SS (*Durie et al., 2020*). Thus, to improve our understanding of the organization of the Dot/Icm T4SS, we used additional cryo-EM data collection and analysis to increase the resolution and quality of the maps, allowing us to build detailed models for the previously unidentified regions of the complex. Here, we report the structure and organization of the *L. pneumophila* Dot/Icm T4SS OMC and PR at resolutions that allow us to identify new components which had not previously been detected by decades of fundamental genetic and biochemical work or by more recent cutting edge cryo-electron tomography. Furthermore, this work reveals how the symmetry mismatch between the OMC and PR is accommodated, an observation that may inform the understanding of symmetry mismatch elements in homologous systems. Additionally, we identified another, unexpected symmetry mismatch between the dome and the rest of the OMC, a feature that has not been observed in other structurally characterized T4SSs and also characterize structural flexibility between the dome and the OMC.

## Results and discussion

### Reconstruction of maps of the Dot/Icm T4SS

We have determined a 3.8 Å asymmetric reconstruction (C1) of the Dot/Icm T4SS using single particle cryo-EM approaches. This resolution made it possible to trace connections between the OMC, which contains 13 copies of the asymmetric unit, and the PR, which contains 18 copies of the asymmetric unit. Unexpectedly, we observed five peptide chains originating from the PR, with attached densities sandwiched between the PR and OMC (*Figure 1A*). Although the resolution was not high enough in the C1 reconstruction to confidently identify the molecular composition of this density, its arrangement in the complex suggested a mechanism for accommodating the symmetry mismatch between the PR and OMC. To increase the resolution of the OMC and PR, C13 and C18 symmetry were applied to the OMC and PR, respectively, in line with our previous report (*Durie et al., 2020*). This extended the resolution to 2.8 Å for both maps (*Figure 1B*). Notably, we were not able to resolve the dome feature in the center of the OMC after the application of symmetry, suggesting that this region contained another symmetry

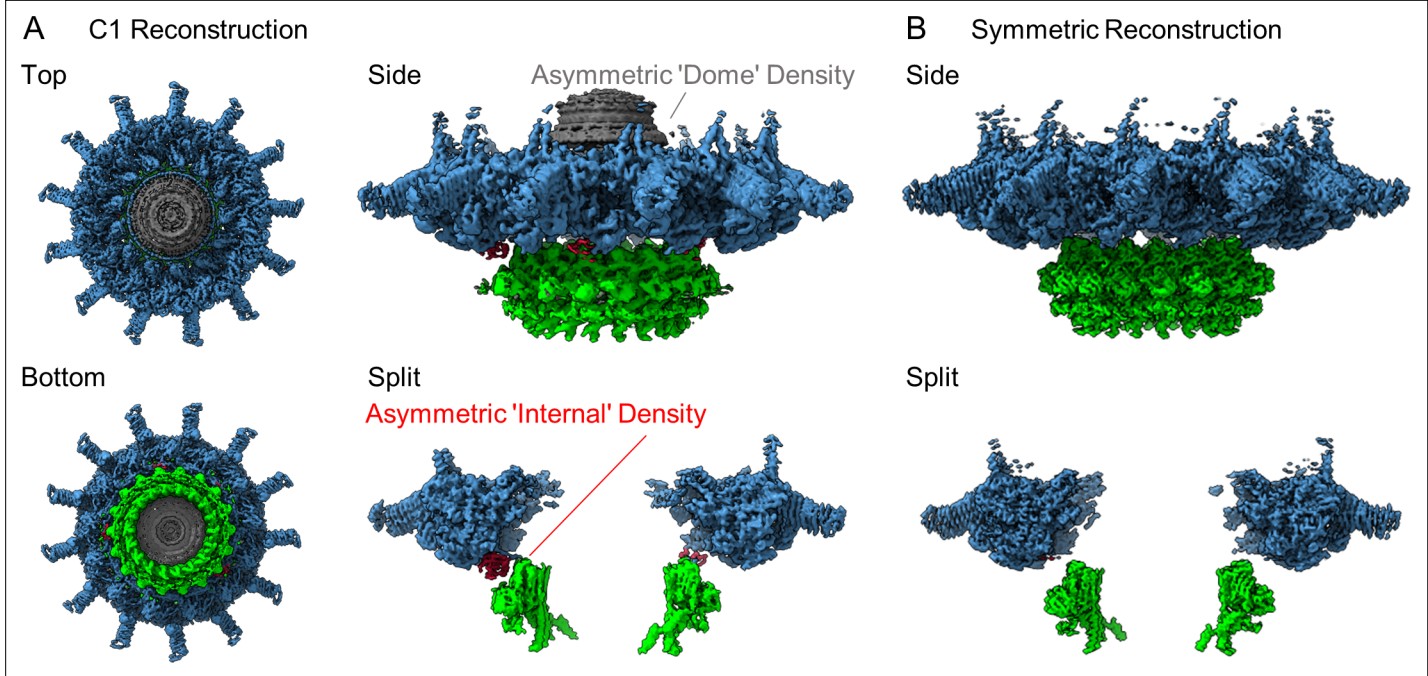

**Figure 1.** The structure of the Dot/Icm type IV secretion system (T4SS). (**A**) The asymmetric reconstruction (**C1**) of the Dot/Icm T4SS includes the outer membrane cap (OMC) (shown in blue), the periplasmic ring (PR) (shown in green), the asymmetric dome (shown in gray), and additional densities with no apparent symmetry sandwiched between the OMC and PR (shown in red). (**B**) The imposition of symmetry was used to improve the resolution of the OMC (C13 symmetry) and PR (C18 symmetry).

The online version of this article includes the following figure supplement(s) for figure 1:

**Figure supplement 1.** Cryo-EM data processing workflow.

**Figure supplement 2.** Cryo-EM data analysis C1 refinements.

**Figure supplement 3.** Cryo-EM data analysis focused refinements.

**Figure supplement 4.** Three-dimensional variability analysis of the Dot/Icm dome.

and/or was structurally flexible. To help better resolve this important region of the map, we implemented a recently developed data analysis strategy, 3D variability analysis (3DVA) (*Punjani and Fleet, 2021*). This computational analysis led to the calculation of five distinct maps of the dome at a global resolution of 4.6 Å, revealing that, unlike the rest of the OMC, the dome is 16-fold symmetrical (*Figure 1—figure supplement 4*). Thus, the Dot/Icm T4SS has three distinct symmetrical regions of the complex, a 16-fold symmetrical dome, a 13-fold symmetrical OMC, and an 18-fold symmetrical PR.

## Architecture of the Dot/Icm T4SS OMC

Models of DotC, DotD$_1$, DotD$_2$, DotH, and DotK were built within the map of the OMC that was reconstructed with C13 symmetry imposed. While parts of these models were presented in our previous report (*Durie et al., 2020*), the quality of the new map of the OMC allowed us to extend many of these models for a nearly complete structural analyses of OMC protein structures and, importantly, to build models within the previously undefined regions of the complex. We now present extended, higher resolution models of DotC (residues 28–35, 60–161, and 173–268), DotD$_1$ (residues 24–162), DotD$_2$ (residues 25–160), DotH (residues 271–361), and DotK (residues 40–188) (*Figure 2* and *Figure 2—figure supplements 1–3 Table 1*). These extended models provide additional insight into the arrangement of the N-termini of these proteins, placing predicted lipidation sites of DotC (C19), DotD$_1$ (C19), DotD$_2$ (C19), and DotK (C27) in positions close to the outer membrane (*Figure 2—figure supplement 4A*; *Yerushalmi et al., 2005*). In addition, we modeled into these maps Lpg0657, which we now call Dis1 (**D**ot/**I**cm **S**ecretion), in close agreement with our previous assignment (*Durie et al., 2020*). However, the model contains a newly resolved extension within the N-terminus of Dis1 (residues 42–65 and 88–98) which forms two helices that extend 'up' from the OMC disk, placing them near, or perhaps within, the outer membrane.

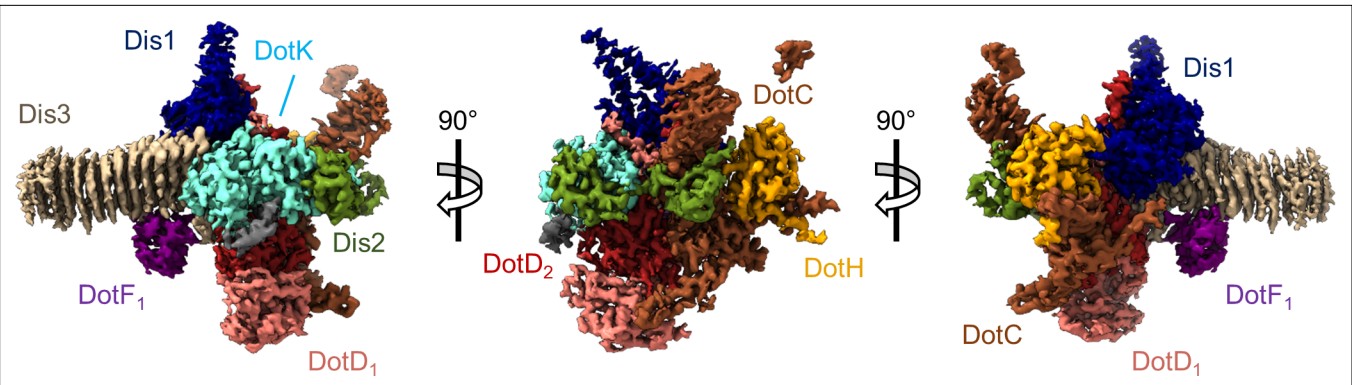

**Figure 2.** The asymmetric unit of the Dot/Icm type IV secretion system (T4SS) outer membrane cap (OMC). The asymmetric unit of the OMC is comprised of one copy of DotC (brown), DotF$_1$ (purple), DotK (cyan), DotH (orange), Dis1 (Lpg0657, blue), Dis2 (Lpg0823, green), Dis3 (Lpg2847, tan), and two copies of DotD (red and pink).

The online version of this article includes the following figure supplement(s) for figure 2:

**Figure supplement 1.** Model statistics for components of the Dot/Icm type IV secretion system (T4SS) outer membrane cap (OMC).

**Figure supplement 2.** Model map correlation for some components of the outer membrane cap (OMC).

**Figure supplement 3.** Model map correlation for remaining components of the outer membrane cap (OMC).

**Figure supplement 4.** Predicted membrane interactions in the outer membrane cap (OMC).

**Figure supplement 5.** Structure of Dis2 (Lpg0823).

**Figure supplement 6.** Structure of Dis3 (Lpg2847).

**Figure supplement 7.** Structure of DotF$_1$ in the outer membrane cap (OMC).

A previous study found that a transposon mutant interrupting the Dis1 gene resulted in a strain that replicates as well as the wild-type strain in liquid culture, but has an intracellular growth defect in both *Acanthamoeba castellanii* and bone marrow-derived murine macrophages, consistent with Dis1 playing an important role in Dot/Icm T4SS function (*Goodwin et al., 2016*).

In addition to the improved models discussed above, this new map identifies the three chains within the OMC that were previously unknown (described simply as 'chain 1', 'chain 2', and 'chain 3') (*Figure 2*). The former 'chain 1' is Lpg0823, now named Dis2 (residues 40–115, *Figure 2—figure supplement 5A*). Dis2 is positioned near the outer membrane and consists of two lobes, similar in both sequence and structure, that are pinned together with a total of five disulfide bonds, all of which show strong density within the map (*Figure 2—figure supplement 5B-D*). A DALI search of Dis2 returned no significant structural similarity among PDB entries, making its function difficult to infer (*Holm, 2020*). Together, the predicted membrane interaction sites in DotC, DotD$_1$, DotD$_2$, DotK, would anchor the Dot/Icm T4SS to the outer membrane at seven positions per asymmetric unit (*Figure 2—figure supplement 4E*). The improved density maps have allowed us to visualize new structural features and details of previously identified Dot/Icm T4SS components and identify the composition of peptide chains that were previously not able to be modeled. The organization of these regions in regard to the location of their predicted lipidation sites provides the first molecular insight into how this complex becomes anchored in the outer membrane of the bacteria.

We identify 'chain 2' as Lpg2847 (residues 29–320), now referred to as Dis3. Dis3, predominantly a β-helical protein, is the major contributor to the 'arms' that extend outward radially from the disk of the OMC (*Figure 2—figure supplement 6A*). Dis3 consists of a total of 14 helical rungs and is structurally similar to membrane associated proteins found in *Parabacteroides distasonis* (gene BDI3087; PDB 3J × 8), *Bacteroides fragilis* (gene BF0425; PDB 3PET), and *B. pertussis* (pertactin; PDB 1DAB, *Figure 2—figure supplement 6B*). The exterior face of Dis3 is predominantly electropositive, potentially facilitating an interaction with the electronegative head groups of the inner leaflet of the outer membrane (*Figure 2—figure supplement 6C*). Notably, the C-terminus of Dis3 contacts three different proteins within the OMC: Dis1, DotK, and DotD$_1$ (*Figure 2—figure supplement 6D*). The presence of Dis2 (Lpg0823, UniprotKB Q5ZXA9) and Dis3 (Lpg2847, UnitprotKB Q5ZRN3) in the isolated Dot/Icm T4SS was confirmed by mass spectrometry using four biologically independent

**Table 1.** Map reconstruction and model refinement.

| | OMC | PR | OMC/PR |
|---|---|---|---|
| EMDB accession codes | 24005 | 24006 | 24004 |
| **Data collection and processing** | | | |
| Magnification | 81,000 × | 81,000× | 81,000× |
| Voltage (kV) | 300 | 300 | 300 |
| Total electron dose (e⁻/Å2) | 50 | 50 | 50 |
| Defocus range (μm) | –1.5 to –2.1 | –1.5 to –2.1 | –1.5 to –2.1 |
| Pixel size (Å) | 1.1 | 1.1 | 1.1 |
| Processing software | CryoSPARC, Relion | CryoSPARC, Relion | CryoSPARC, Relion |
| Initial particles (number) | 1,389,426 | 1,389,426 | 1,389,426 |
| Final particles (number) | 84,886 | 43,907 | 136,818 |
| Map sharpening B-factor | –99.2 | –82.8 | –115 |
| Map resolution (Å) | 2.8 | 2.8 | 2.8 |
| FSC threshold | 0.143 | 0.143 | 0.143 |
| **Model refinement and validation** | | | |
| Starting models | 6 × 62 | 6 × 64 | 6 × 65 |
| **FSC** | | | |
| 0.5 | 2.8 | 2.8 | 4.0 |
| 0.143 | 2.7 | 2.7 | 3.8 |
| **Model residues** | | | |
| Total | 17,264 | 5490 | 23.286 |
| DotC | 28–35, 60–161, 173–268 | – | 28–35, 57–161, 173–272 |
| DotD$_1$ | 24–162 | – | 23–162 |
| DotD$_2$ | 25–160 | – | 24–160 |
| DotF$_1$ | 208–266 | | 208–266 |
| DotF$_2$ | – | 207–269 | 207–269 |
| DotG | – | 791–824 | 791–824 |
| DotH | 271–361 | 104–263 | 104–361 |
| DotK | 40–188 | – | 38–188 |
| Dis1 | 42–65, 88–234 | – | 42–65, 88–236 |
| Dis2 | 40–115 | – | 39–116 |
| Dis3 | 29–320 | – | 29–320 |
| Unknown OMC | 122–130 | – | 1–54, 122–130 |
| Unknown PR | – | 1–38, 70–79 | 1–38, 70–79 |
| Clashscore | 3.04 | 0.67 | 5.80 |
| Molprobity | 1.42 | 0.85 | 2.22 |
| **Bonds** | | | |
| Lengths (Å) | 0.003 | 0.004 | 0.012 |
| Angles (°) | 0.748 | 0.785 | 1.822 |

*Table 1 continued on next page*

*Table 1 continued*

| | OMC | PR | OMC/PR |
|---|---|---|---|
| Ramachandran (%) | | | |
| Favored | 95.3 | 97.3 | 94.5 |
| Allowed | 4.7 | 2.7 | 5.5 |
| Outliers | 0.0 | 0.0 | 0.0 |
| B-factors | 55.3 | 69.7 | 165.0 |
| PDB | 7MUD | 7MUE | 7MUC |

purifications (*Table 2*). Lpg2847 was previously identified as part of a two-gene operon with lpg2848 or snrnA, an RNase secreted by the *L. pneumophila* T2SS (*Rossier et al., 2009*). Insertional mutants of both the srnA and dis1 genes were prepared and tested for the ability to secrete the RNase through the T2SS (*Rossier et al., 2009*). The dis1 mutant showed no growth defect in the protozoan host *H. vermiformis*, a well-accepted assay for T2SS function, but has not yet been tested specifically for T4SS assembly or function (*Rossier et al., 2009*).

This is not the first time that additional components have been described in the Dot/Icm T4SS directly from the cryo-EM density map. In fact, in 2020 alone three proteins have been identified as components of the Dot/Icm T4SS through high-resolution cryo-EM: Lpg0294 (DotY), Lpg0657 (Dis1), and Lpg1549 (DotZ) (*Durie et al., 2020*; *Meir et al., 2020*). Although there is no apparent link between these three genes and the other known components of the Dot/Icm T4SS, a targeted genetic analysis may shed light on how these genes have been integrated into this system. Further studies are needed to probe the function of these newly identified Dot/Icm components.

Finally, we identified 'chain 3' as DotF (DotF$_1$, residues 208–266). This portion of DotF consists of a small globular fold comprised of two β-sheets each consisting of three strands (*Figure 2—figure supplement 7A*). One face of DotF interacts with rungs 9, 10, and 11 of Dis3 (*Figure 2—figure supplement 7B*). The interface between the two proteins is comprised of hydrogen bonds, hydrophobic interactions, and salt bridges (*Figure 2—figure supplement 7C*). A DALI search of this portion of DotF resulted in several hits, most notably a T2SS protein from *Escherichia coli* known as GspC (PDB 3OSS) and a pilus protein from *Pseudomonas aeruginosa* known as PilP (PDB 21C4, *Figure 2—figure supplement 7D*). Like DotF, both structurally similar proteins are found within the periplasm in their respective systems (*Korotkov et al., 2011*; *Tammam et al., 2011*).

## Architecture of the Dot/Icm T4SS PR

There are a total of four chains within the C18 symmetry-imposed map of the PR. Three of these chains were unambiguously determined to be portions of a second copy of DotF (DotF$_2$), DotG, and DotH, while one chain could not be identified and was left as a polyalanine chain model (*Figure 3* and *Figure 3—figure supplements 1–2* ). Located in the interior of the PR is a portion of DotG (residues 791–824) consisting of a single helix that starts from the inner membrane side of the PR and is followed by a short loop that extends toward the outer membrane or 'top' of the complex (*Figure 3—figure supplement 3A*). The short loop of DotG contacts a globular domain of DotH consisting of two β-sheets composed of four and five β-strands that contain residues 104–263 (*Figure 3—figure supplement 3B*). The interaction that is observed between DotG/DotH in the PR is similar in structure to those previously reported for VirB10/VirB9 and CagX/CagY and likely reflects an important function for retaining this organization within the PR (*Sheedlo et al., 2020*; *Sgro et al., 2018*).

The structure of DotH is similar to the N-terminal regions of VirB9 (PDB 6GYB, residues 27–133) and CagX (PDB 6 × 6 J, residues 32–311), making DotH a bona fide structural homolog of VirB9 (*Figure 3—figure supplement 3C*,D) despite very little sequence similarity (*Sheedlo et al., 2020*; *Sgro et al., 2018*). The arrangement of DotH and DotG within the PR is similar to interactions between VirB9/VirB10 (PDB 6GYB) in the *Xanthomonas citri* T4SS and CagX/CagY (PDB 6X6J ) within the PR of the *H. pylori* Cag T4SS (*Figure 3—figure supplement 3E*; *Sheedlo et al., 2020*; *Sgro et al., 2018*). Interestingly, the *X. citri* T4SS core complex is smaller and composed of fewer components than the Dot/Icm and Cag T4SSs, and its region that shows structural similarity to the PR of the Dot/Icm and Cag

**Table 2.** Dot/Icm proteins in isolated complex sample.

| Identified proteins | Gene number* | Spectral counts[†] | | | |
|---|---|---|---|---|---|
| | | Prep 1 | Prep 2 | Prep 3 | Prep 4 |
| **DotG** [‡] | Q5ZYC1 | 112 | 114 | 195 | 150 |
| **DotF** | Q5ZYC0 | 94 | 69 | 101 | 107 |
| DotA | Q5ZS33 | 38 | 65 | 60 | 69 |
| **Dis3** | Q5ZRN3 | 22 | 24 | 41 | 87 |
| DotO | Q5ZYB6 | 37 | 38 | 47 | 27 |
| **DotH** | Q5ZYC2 | 28 | 19 | 28 | 47 |
| IcmF | Q5ZYB4 | 15 | 18 | 36 | 38 |
| IcmX | Q5ZS30 | 19 | 13 | 28 | 33 |
| DotL | Q5ZYC6 | 10 | 26 | 20 | 32 |
| DotD | Q5ZS45 | 11 | 9 | 14 | 33 |
| **DotC** | Q5ZS44 | 9 | 11 | 16 | 22 |
| DotB | Q5ZS43 | 16 | 11 | 7 | 12 |
| **Dis1** | Q5ZXS4 | 6 | 4 | 16 | 19 |
| DotY | Q5ZYR7 | 4 | 9 | 4 | 10 |
| DotM | Q5ZYC7 | 2 | 14 | 3 | 9 |
| **DotK** | Q5ZYC5 | 2 | 6 | 10 | 9 |
| IcmW | Q5ZS31 | 5 | 7 | 5 | 8 |
| DotZ | Q5ZV91 | 1 | 8 | 3 | 9 |
| DotN | Q5ZYB7 | 2 | 6 | 4 | 7 |
| **Dis2** | Q5ZXA9 | 2 | 4 | 3 | 5 |
| DotI | Q5ZYC3 | 1 | 3 | 5 | 3 |
| IcmV | Q5ZS32 | 1 | 1 | 2 | 7 |
| IcmT | Q5ZYD1 | 1 | 1 | 3 | 1 |
| IcmS | Q5ZYD0 | | 4 | 1 | 2 |

*UniProtKB Accession Number.

[†]Proteins were identified by searching the MS/MS data against L. pneumophila (UniProt; 2930 entries) using Proteome Discoverer (v2.1, Thermo Scientific). Search parameters included MS1 mass tolerance of 10 ppm and fragment tolerance of 0.1 Da. False discovery rate (FDR) was determined using Percolator and proteins/peptides with a FDR of ≤1% were retained for further analysis. Complete results are in Supplementary file 1 in supplementary material.

[‡]Components identified in this structure are in bold.

T4SSs has previously not been considered a separate region of the *X. citri* core complex. Instead, previous reports describe an outer and inner layer of the *X. citri* T4SS (*Sgro et al., 2018*). However, when comparing the structures of the Dot/Icm and Cag T4SSs with the prototype *X. citri* T4SSs, it appears that its inner layer should now be considered structurally similar to the PR regions of the larger Dot/Icm and Cag T4SSs. Importantly, one major distinction observed for the *X. citri* T4SS is that the outer and inner layers of its core complex share the same symmetry operator (C14), rather than contain a symmetry mismatch as observed for both the Dot/Icm and Cag T4SSs.

Located adjacent to DotH and on the periphery of the PR is a small globular domain that we identified as DotF (residues 207–269). This portion of DotF consists of the same residues as was discovered in the OMC (DotF$_1$) and thus, likely represents unique copies of DotF which we call DotF$_2$. DotF$_2$ is nearly identical in structure to the model of DotF$_1$ that was built in the C13 symmetry-imposed maps of the OMC (RMSD of 0.5 Å). This results in a total of 31 copies of DotF contained within the intact Dot/Icm T4SS (*Figure 3—figure supplement 4A*). DotF$_2$ engages DotH using a similar interface to that of DotF$_1$ and Dis3, with buried surface areas of 575 and 655 Å, respectively (*Figure 3—figure supplement 4B-D*).

## Models constructed within the asymmetric map

The models that were generated from the 2.8 Å resolution, symmetry-imposed reconstructions of the OMC and PR were fit into the 3.8 Å map of the Dot/Icm T4SS that was generated without the imposition of symmetry. All models fit well within the asymmetric map with only minor changes in the positions of backbone atoms observed for each protein (*Figure 4* and *Figure 4—figure supplements 1–5*). Notably, we modeled portions of two regions of the map that contained additional density within the asymmetric reconstruction. First, we observe 13 linkers between the OMC and PR (identified as residues 264–270 of DotH) in various conformations, revealing the direct connections between the two regions. Second, we observed five small globular folds located between the OMC and PR (*Figure 4A*, *Figure 4—figure supplement 6A-C*). These folds consist of two β-sheets and incorporate the linker from DotH as an additional β-strand in one of the two sheets (*Figure 4—figure supplement 6C*). Close inspection clearly showed that these domains contain folds nearly identical to that of the C-terminal domain of DotH in the OMC (residues 278–360). Upon fitting the C-terminal domain of DotH into this portion of the map, the register correlates well with the model, with mean side-chain CC values ranging from 0.68 to 0.72 (*Figure 4—figure*

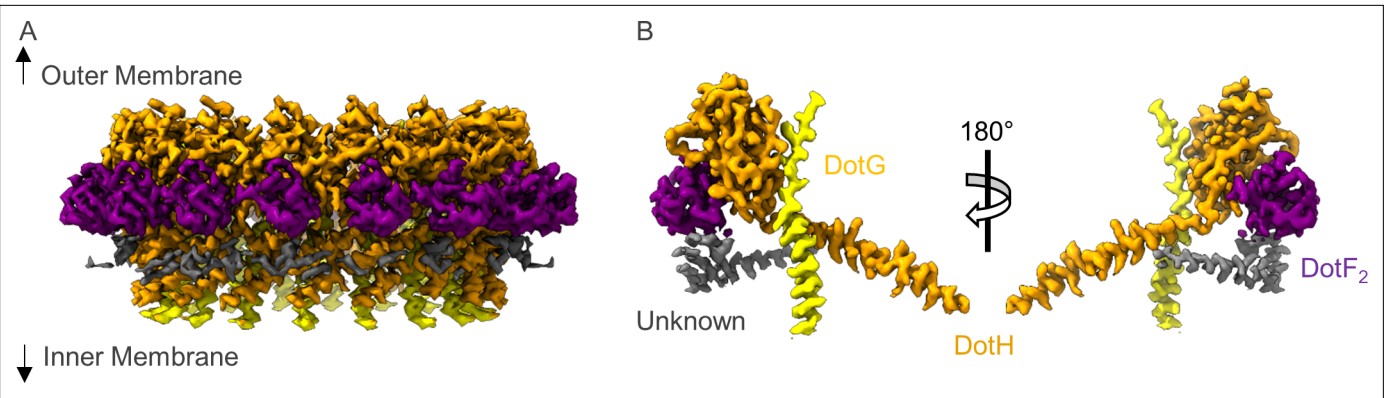

**Figure 3.** The asymmetric unit of the periplasmic ring (PR). (**A**) The PR of the Dot/Icm type IV secretion system (T4SS) is comprised of at least four peptide chains. Three have been identified as DotG (yellow), DotH (orange), and DotF$_2$ (purple). A fourth chain could not be identified and is shown in gray. (**B**) Within the asymmetric unit, we observe interactions between DotF$_2$-DotH and DotH-DotG.

The online version of this article includes the following figure supplement(s) for figure 3:

**Figure supplement 1.** Model statistics for components of the periplasmic ring (PR).

**Figure supplement 2.** Model map correlation for identified components in the periplasmic ring (PR).

**Figure supplement 3.** Organization of components within the periplasmic ring (PR).

**Figure supplement 4.** The structure of DotF$_2$ in the periplasmic ring (PR).

*supplement 6D*). We propose that these five additional domains of the DotH C-terminal domain are the five copies that do not span the symmetry mismatch between the OMC and PR. Thus, there are 18 copies of DotH in the entire structure with all 18 NTDs comprising the PR, 13 of the associated CTDs extending up to build part of the OMC disk, and the other five CTDs extending up only part way into the intervening space. Each of the intervening DotH C-terminal domains between the OMC and PR occurs every two to three asymmetric units, and the degree to which they can be observed varies,

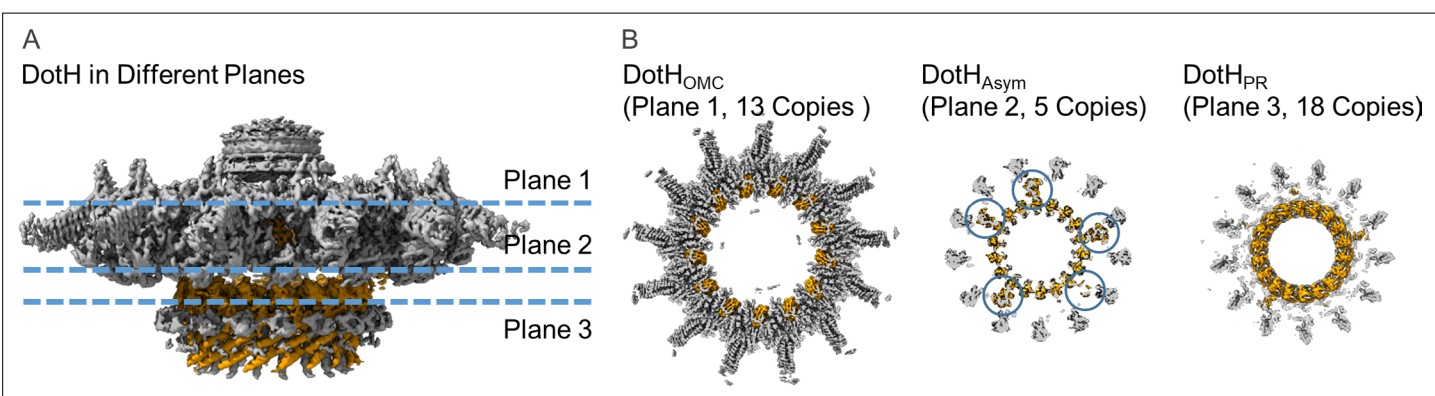

**Figure 4.** Organization of the DotH C-terminal domain located between the outer membrane cap (OMC) and periplasmic ring (PR). (**A**) Within the asymmetric reconstruction of the Dot/Icm type IV secretion system (T4SS), we observe DotH (orange) in three separate places, the OMC (plane 1), the PR (plane 3), and the space in between the two (plane 2). (**B**) The number of copies of DotH in each plane differs. There are 13 symmetrical copies in the OMC (plane 1), 5 asymmetrical copies in the space between the OMC and PR (plane 2), and 18 symmetrical copies in the PR (plane 3).

The online version of this article includes the following figure supplement(s) for figure 4:

**Figure supplement 1.** Model map Fourier shell correlation (FSC) of each chain in the asymmetric reconstruction.

**Figure supplement 2.** Model map Fourier shell correlation (FSC) of each chain in the asymmetric reconstruction continued.

**Figure supplement 3.** Model map correlation for some of the identified components in the C1 reconstruction.

**Figure supplement 4.** Model map correlation for some of the identified components in the C1 reconstruction.

**Figure supplement 5.** Model map correlation for some of the identified components in the C1 reconstruction.

**Figure supplement 6.** Variation among copies of DotH.

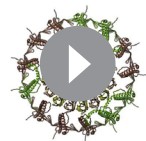

**Video 1.** Positions of the DotH N-terminal and C-terminal domains within the asymmetric reconstruction of the Dot/Icm type IV secretion system (T4SS). Within the asymmetric reconstruction of the Dot/Icm T4SS we observe DotH (colored in orange) in three distinct portions: the outer membrane cap (OMC), the periplasmic ring (PR), and the asymmetric region between the OMC and PR. The region of DotH observed in each portion varies with the N-terminal domain observed only in the PR and the C-terminal domain observed in the OMC and the asymmetric region. The number of copies of DotH also varies by position with 18 copies of the N-terminal domain observed in the PR, 5 copies of the C-terminal domain observed in the asymmetric region, and 13 copies of the C-terminal domain observed in the OMC.
https://elifesciences.org/articles/70427/figures#video1

**Video 2.** The extent to which the DotC N-terminal extension varies among maps reconstructed using 3D variability analysis. Within the five maps that were used to resolve the dome region in the outer membrane cap (OMC) we note that the extent to which the DotC N-terminus varies considerably. The extension of DotC is only strongly observed in four or five copies of DotC in each map.
https://elifesciences.org/articles/70427/figures#video2

indicating that the position of these domains is not static with respect to the PR or the OMC (*Figure 4B* and *Video 1*). The flexibility of these five DotH CTDs suggests a mechanism through which the symmetry mismatch observed between the OMC and PR can be accommodated though the utility of the symmetry mismatch cannot be inferred.

Interestingly, we do not see similar densities between the OMC and PR of the *H. pylori* Cag T4SS, even though there is also a symmetry mismatch in this system and DotH is structurally homolgous to CagX. With this understanding of how the symmetry mismatch is accommodated in the Dot/Icm T4SS, we propose that flexible and/or dynamic connection between regions of PR and OMC, that are not seen in the Cag T4SS, will be important for the Dot/Icm T4SS translocating such a uniquely large repertoire of secretion substrates. Although we are now in a position to describe how the symmetry mismatch is accommodated, its impact on function remains to be determined.

## The dome density contains the C-terminus of DotG

Five maps of the T4SS were reconstructed using a 3DVA in cryoSPARC that resolved the secondary structure of the dome positioned in the center of the OMC (*Figure 5A*, and *Table 3*). To conduct this analysis, the particles had to be downsampled from 1.1 to 2.2 Å/pix, resulting in a lower global resolution of ~4.6 Å. Within this dome there are 16 α-helices that appear nearly identical at this resolution. Since this resolution is already close to the Nyquist limit of the data (~4.4 Å), imposing C16 symmetry did not improve the resolution of the maps. However, based on previous studies of T4SSs, we reasoned that this portion of the T4SS may correspond to DotG, the proposed *L. pneumophila* homolog of VirB10 via sequence comparisons (*Nagai and Kubori, 2011*). Indeed, a model of the C-terminal domain of DotG generated in Swiss Model using VirB10 as a template fits into the resolved dome density (*Figure 5B*; *Waterhouse et al., 2018*). This finding is consistent with our previous observation that the Dot/Icm T4SS core complex isolated from a *dotG* deletion strain lacked the dome portion of the OMC (*Durie et al., 2020*). Interestingly, the interface between the 16 copies of DotG and the rest of the OMC disk is sparse, potentially leading to the low-resolution reconstruction of this portion of the map (*Video 2*). Based on the structure of DotG modeled within the PR (residues 791–824) and the homology model fit into the dome density (consisting approximately of residues 857–1046), we hypothesize that DotG extends from the PR to the dome, though a physical connection between the two structures is not observed. This would lead to a total of 18 copies of DotG in the intact Dot/Icm T4SS with two copies of DotG not visualized within the dome likely due to structural

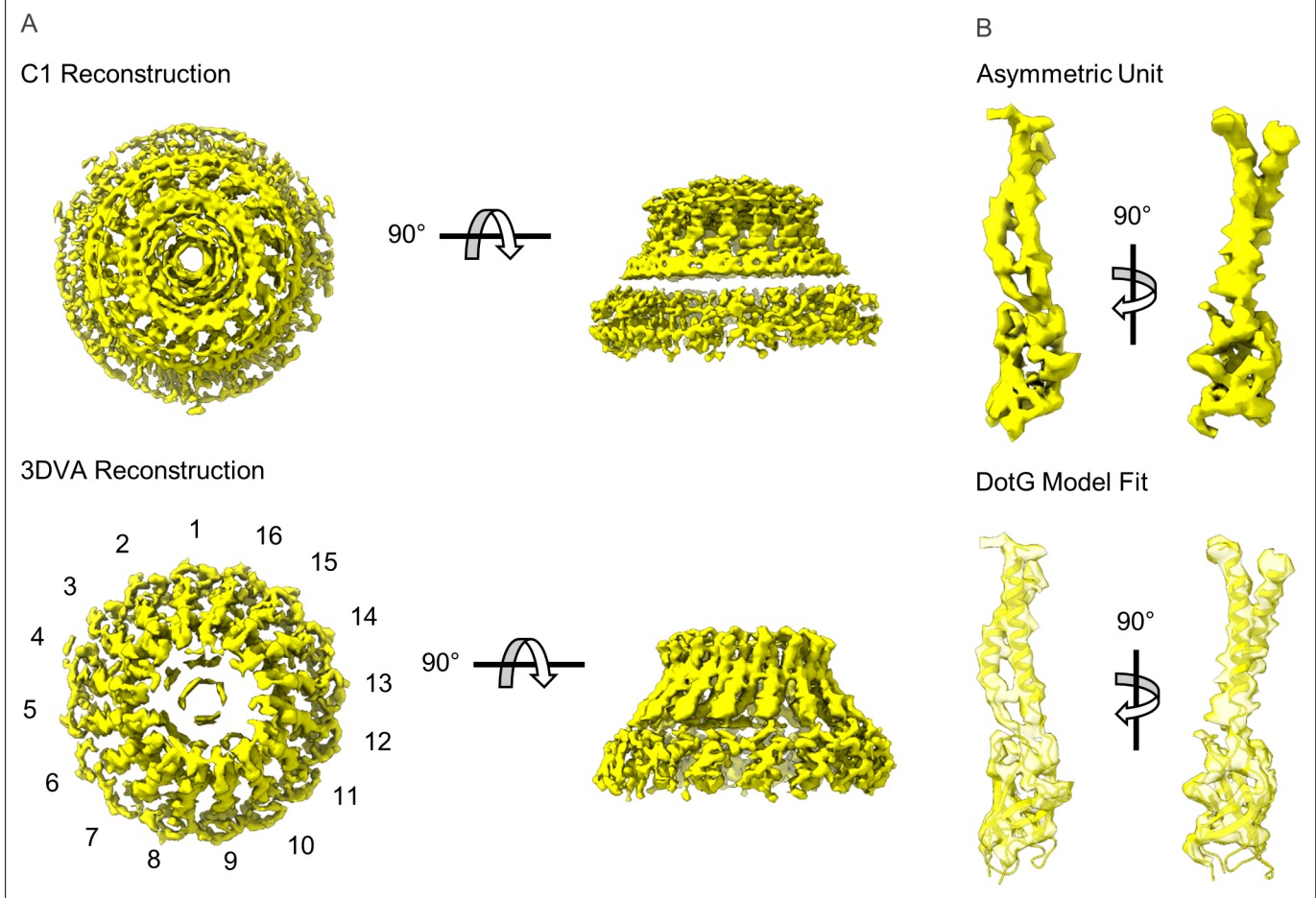

**Figure 5.** Modeling DotG within the 'dome' density. (**A**) The 'dome' density located in the center of the outer membrane cap (OMC) was uninterpretable in the C1 reconstruction (top), likely due to structural heterogeneity. The density is considerably improved after 3D variability analysis with an apparent 16-fold symmetry observed about the middle of the structure (bottom). (**B**) The density within the asymmetric unit of the dome contains similar structural features seen in VirB10 homologs (top). A model of the DotG C-terminus was constructed in Swiss Model which fits into this density after refinement (bottom).

The online version of this article includes the following figure supplement(s) for figure 5:

**Figure supplement 1.** Arrangement of the DotC N-terminus in the maps reconstructed using 3D variability analysis.

heterogeneity. Taken together, we propose a final stoichiometry of 31:26:18:18:13:13:13:13:13 (DotF:DotD:DotG:DotH:DotC:DotK:Dis1:Dis2:Dis3) for the Dot/Icm T4SS.

In addition to the portion of the dome that we predict is DotG, there are also two segments of DotC that were not resolved in our previous electron microscopy density maps. These include an internal connection (residues 162–172) and a relatively long N-terminal extension (residues 28–57, *Figure 5*, *Figure 5—figure supplement 1*). The internal bridge within DotC consists of a single loop connecting residues 161 and 173. In contrast, the N-terminal extension of DotC was modeled as two relatively large helices positioned near DotG and bridging adjacent asymmetric units (*Figure 5— figure supplement 1B*). Notably, this portion of DotC was only observed in either four or five of the 13 copies of DotC observed in each map. The copies of DotC that contain this portion are positioned such that this extension is in a similar position relative to DotG within the dome (*Figure 5—figure supplement 1B* and *Video 3*).

The discovery of the DotG C-terminal domain within the dome density is in line with previous reports that hypothesized that DotG is a homolog of VirB10. However, we have unexpectedly uncovered that the Dot/Icm incorporates only 16 copies of DotG into the OMC dome out of the 18 DotG copies in the PR, giving rise to another symmetry mismatch within this system. The finding that there is variability within the N-terminus of DotC also suggests that interactions between DotG and the rest

**Table 3.** Three-dimensional variability analysis (3DVA) map reconstruction and model refinement.

| | Map 1 | Map 2 | Map 3 | Map 4 | Map 5 |
|---|---|---|---|---|---|
| EMDB accession codes | 24018 | 24020 | 24023 | 24024 | 24026 |
| Data collection and processing | | | | | |
| Magnification | 81,000 × | 81,000× | 81,000× | 81,000× | 81,000× |
| Voltage (kV) | 300 | 300 | 300 | 300 | 300 |
| Total electron dose (e⁻/Å²) | 50 | 50 | 50 | 50 | 50 |
| Defocus range (µm) | −1.5 to −2.1 | −1.5 to −2.1 | −1.5 to −2.1 | −1.5 to −2.1 | −1.5 to −2.1 |
| Pixel size (Å) | 1.1 | 1.1 | 1.1 | 1.1 | 1.1 |
| Processing software | CryoSPARC | CryoSPARC | CryoSPARC | CryoSPARC | CryoSPARC |
| Initial particles (number) | 303,447 | 303,447 | 303,447 | 303,447 | 303,447 |
| Final particles (number) | 79,720 | 75,959 | 64,698 | 42,013 | 41,057 |
| Map sharpening B-factor | −88.5 | −85.4 | −85,8 | −84.4 | −80.1 |
| Map resolution (Å) | 4.6 | 4.6 | 4.6 | 4.6 | 4.6 |
| FSC threshold | 0.143 | 0.143 | 0.143 | 0.143 | 0.143 |
| Model refinement and validation | | | | | |
| Starting models | 7MUC | 7MUC | 7MUC | 7MUC | 7MUC |
| FSC | | | | | |
| 0.5 | 4.6 | 4.6 | 4.6 | 4.8 | 4.8 |
| 0.143 | 4.5 | 4.5 | 4.5 | 4.5 | 4.5 |
| Model residues | | | | | |
| Total | 25,121 | | | | |
| DotC | 59–267 | | | | |
| DotD$_1$ | 23–162 | | | | |
| DotD$_2$ | 24–160 | | | | |
| DotF$_1$ | 208–266 | | | | |
| DotF$_2$ | 207–269 | | | | |
| DotG | 791–824, 862–978, 999–1046 | | | | |
| DotH | 104–361 | | | | |
| DotK | 38–188 | | | | |
| Lpg0657 | 42–65, 88–236 | | | | |
| Lpg0823 | 39–116 | | | | |
| Lpg2847 | 113–320 | | | | |
| Unknown OMC | 122–130 | | | | |
| Unknown PR | 1–38, 70–79 | | | | |
| Clashscore | 9.55 | 9.64 | 15.53 | 9.47 | 9.85 |
| Molprobity | 2.80 | 2.77 | 2.76 | 2.68 | 2.73 |
| Bonds | | | | | |
| Lengths (Å) | 0.006 | 0.005 | 0.003 | 0.004 | 0.004 |
| Angles (°) | 0.798 | 0.765 | 0.540 | 0.702 | 0.711 |

*Table 3 continued on next page*

*Table 3 continued*

|  | Map 1 | Map 2 | Map 3 | Map 4 | Map 5 |
|---|---|---|---|---|---|
| Ramachandran (%) |  |  |  |  |  |
| Favored | 94.0 | 94.2 | 94.7 | 95.2 | 94.7 |
| Allowed | 6.0 | 5.8 | 5.3 | 4.8 | 5.3 |
| Outliers | 0.0 | 0.0 | 0.0 | 0.0 | 0.0 |
| B-factors | 147.96 | 154.48 | 146.18 | 145.88 | 145.07 |
| PDB | 7MUQ | 7MUS | 7MUV | 7MUW | 7MUY |

of the OMC disk are primarily mediated by DotC. Future studies will seek to resolve the origin of the C16:C18 (dome:PR) mismatch, including the location and role of the two DotG C-terminal domains that do not span the mismatch, as well as the interactions between DotC and DotG.

When comparing the five maps determined using 3DVA, not only does DotG sample different positions in the 'dome' with respect to the rest of the OMC, but the PR occupies different positions with respect to the OMC disk as well. Having noted that the OMC disk is anchored into the outer membrane with as many as seven interactions per asymmetric unit (or over 90 for the complete complex) and having observed multiple conformations of the OMC dome and the PR relative to the OMC disk, we visualized the continuously distributed conformations in the context of a movie (*Video 4*). The movie shows that the OMC dome, OMC disk, and PR of Dot/Icm T4SS can accommodate various orientations in relation to each other that suggest that the complex could undergo a ratcheting motion, in which the dome and PR rotate back and forth about the central axis with the different regions of the complex tethered together by the physical connections across the symmetry mismatches. This leads to a prediction that the multiple symmetry mismatches and various conformational states of the complex are important in how the Dot/Icm T4SS accommodates a larger number of protein substrates than other T4SSs (*Schroeder, 2017*).

# Materials and methods

**Key resources table**

| Reagent type (species) or resource | Designation | Source or reference | Identifiers | Additional information |
|---|---|---|---|---|
| Strain, strain background (*Legionella pneumophila*) | Lp02; WT | PMID:23717549 |  |  |
| Software, algorithm | MotionCor2 | PMID:28250466 |  |  |
| Software, algorithm | CTFFind4 | PMID:26278980 |  |  |
| Software, algorithm | cryoSPARC | PMID:28165473 PMID:33582281 |  |  |
| Software, algorithm | RELION | PMID:27685097 PMID:30412051 |  |  |
| Software, algorithm | Coot | PMID:20383002 |  |  |
| Software, algorithm | UCSF Chimera | PMID:15264254 PMID:29340616 |  |  |
| Software, algorithm | PHENIX | PMID:29872004 |  |  |
| Software, algorithm | DALI server | PMID:31263867 PMID:31606894 |  |  |

## Preparation of strains

*L. pneumophila* was cultured in ACES (Sigma)-buffered yeast extract broth at pH 6.9 supplemented with 0.1 mg/ml thymidine, 0.4 mg/ml L-cysteine, and 0.135 mg/ml ferric nitrate or on solid medium of this

broth supplemented with 15 g/l agar and 2 g/l charcoal. The *L. pneumophila* laboratory strain Lp02, a thymidine auxotroph derived from the clinical isolate Philadelphia-1 (*Rao et al., 2013*), was utilized.

## Complex isolation

Complexes were isolated from wild-type *L. pneumophila* strain Lp02 as described (*Durie et al., 2020*; *Kubori et al., 2014*; *Kubori and Nagai, 2019*). Cells were suspended in 140 ml of buffer containing 150 mM Trizma base pH 8.0, 500 mM NaCl, and EDTA-free Complete Protease Inhibitor (Roche) at 4 °C. The suspension was incubated on the benchtop, with stirring, until it reached ambient temperature. PMSF (final concentration 1 mM), EDTA (final concentration 1 mM), and lysozyme (final concentration 0.1 mg/ml) were added, and the suspension was incubated at ambient temperature for an additional 30 min. Bacterial membranes were lysed using detergent and alkaline lysis. Triton X-100 (20% w/v) with AG501-X8 resin (BioRad) was added dropwise, followed by $MgSO_4$ (final concentration 3 mM), DNaseI (final concentration 5 µg/ml), and EDTA (final concentration 10 mM), and then the pH was adjusted to 10.0 using NaOH. The remaining steps were conducted at 4 °C. The cell lysate was subjected to centrifugation at 12,000 × $g$ for 20 min to remove unlysed material. The supernatant was then subjected to ultracentrifugation at 100,000 × $g$ for 30 min to pellet membrane complexes. The membrane complex pellets were resuspended and soaked overnight in a small volume of TET buffer (10 mM Trizma base pH 8.0, 1 mM EDTA, 0.1 % Triton X-100). The resuspended sample was then subjected to centrifugation at 14,000 × $g$ for 30 min to pellet debris. The supernatant was subjected to ultra-centrifugation at 100,000 × $g$ for 30 min. The resulting pellet was resuspended in TET and complexes were further separated by Superose 6 10/300 column chromatography in TET buffer with 150 mM NaCl using an AKTA Pure system (GE Life Sciences). The sample collected from the column was used for microscopy. Mass spectrometry analysis was performed as described (*Anwar et al., 2018*).

## Cryo-EM data collection and map reconstruction

For cryo-EM, 4 µl of the isolated Dot/Icm T4SS sample was applied to a glow discharged ultrathin continuous carbon film on Quantifoil 2/2 200 mesh copper grids (Electron Microscopy Services). The sample was applied to the grid 5 consecutive times and incubated for ~60 s after each application. The grid was then rinsed in water to remove detergent before vitrification by plunge-freezing in a slurry of liquid ethane using an FEI vitrobot at 4 °C and 100 % humidity.

The data were collected at the Stanford-SLAC Cryo-EM Facility (Menlo Park, CA) using Titan Krios microscopes (Thermo Fisher, Waltham, MA) operated at 300 keV and equipped with a Quantum energy filter. The images were collected with a K3 Summit direct electron detector operating in counting mode, at a nominal magnification of 81,000, corresponding to a pixel size of 1.1 Å. The energy slit was set at a width of 15 eV. The total dose was 50 e/Å², fractionated over 33 frames in 2.96 s. Data were collected using EPU software

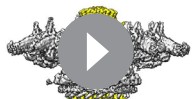

**Video 3.** Positions of portions of DotG found within the map 1 of the 3D variability reconstruction of the Dot/Icm type IV secretion system (T4SS). Within the dome density we observe 16 helical protrusions that contain a fold similar to that of VirB10. These 16 folds are positioned directly above the 18 helices that we have identified as DotG in the periplasmic ring (PR). It is suspected that the heterogeneity of DotG within the disk arises from the sparse contacts observed between DotG and DotC along with a flexible linker that appears to connect the two segments of DotG shown here.
https://elifesciences.org/articles/70427/figures#video3

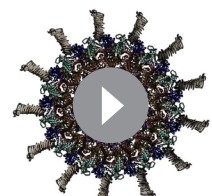

**Video 4.** Maps reconstructed using 3D variability analysis of the Dot/Icm type IV secretion system (T4SS). Using 3D variability analysis, we reconstructed a total of five maps that displayed differences in the way the outer membrane cap (OMC) and periplasmic ring (PR) were positioned. These maps led to the identification of an approximate 16-fold symmetry about the center of the dome positioned within the OMC.
https://elifesciences.org/articles/70427/figures#video4

(Thermo Fisher, Waltham, MA) with a nominal defocus range set from −1.5 to −2.1 μm. A total of 12,263 micrographs were collected.

The video frames were first dose-weighted and aligned using Motioncor2 (*Zheng et al., 2017*). The contrast transfer function (CTF) values were determined using CTFFind4 (*Rohou and Grigorieff, 2015*). Image processing was carried out using cryoSPARC, RELION 3.0, and RELION 3.1 (*Punjani et al., 2017*; *Zivanov et al., 2018*). Using the template picker in cryoSPARC, 1,389,426 particles were picked from 12,204 micrographs. Particles were extracted using a 510 pixel box size (1.1 Å/pix). The extracted particles were used to generate representative 2D classes in cryoSPARC and ~136,000 particles found in the well-resolved classes were kept. The selected particles were used for an ab initio model in cryoSPARC, which was then used as the reference for 3D auto-refinement with and without C13 symmetry (lowpass filtered to 30 Å). Finally, a solvent mask and B-factor were applied to improve the overall features and resolution of the 3D maps with and without C13 symmetry, resulting in reconstruction of 3D maps with a global resolution of 3.4 Å (C13) and 3.8 Å (C1).

The C13 refined volumes and corresponding particles were then exported to RELION for focused refinements. Estimation of beam-tilt values (CTF-refinement) was applied to the selected particles using RELION. With the CTF-refined particle stack, C13 symmetry-imposed refinement with a soft mask around the core complex was done, resulting in a 3.8 Å resolution 3D map.

For focused refinement of the OMC disk, signal subtraction for each particle containing the OMC disk was used with a soft mask. The subtracted particles were subjected to alignment-free focused 3D classification (three classes). The best resolved 3D class of the OMC (~89,000 particles) was then subjected to a masked 3D refinement with local angular searches using C13 symmetry resulting in a 3.7 Å resolution density map. Estimation of per-particle defocus values (CTF-refinement) was applied to the selected particles using RELION. With the CTF-refined particle stack, C13 symmetry-imposed refinement with a soft mask around the OMC disk region of the Dot/Icm T4SS core complex was done, resulting in a 3.2 Å resolution 3D map that contained improved structural features. B-factor sharpening and calculation of masked Fourier shell correlation (FSC) curves steps resulted in the final OMC disk map with 2.8 Å resolution.

The same steps were followed for focused refinement of the PR, starting with signal subtraction for each particle containing the PR with a soft mask. The subtracted particles were subjected to alignment-free focused 3D classification (three classes). The best resolved 3D class of the PR (~44,000 particles) was selected based on class distribution (particle distribution), estimated resolution, and comparison of the 3D density maps. This class was then subjected to a masked 3D refinement with local angular searches using C18 symmetry resulting in a 7.5 Å resolution. Estimation of per-particle defocus values (CTF-refinement) was applied to the selected particles using RELION. With the CTF-refined particle stack, C18 symmetry-imposed refinement with a soft mask around the PR region of the Dot/Icm T4SS core complex was done, resulting in a 4.1 Å resolution 3D map. These maps contained improved features compared to those prior to CTF-refinement. B-factor sharpening and calculation of masked FSC curves resulted in the final PR map with 2.8 Å resolution in.

In a separate workflow from that used for focused refinements, 3DVA in cryoSPARC was performed to assess continuous flexibility in the dome of the OMC. For this analysis, ~136,000 particles, which had resulted in a 3.8 Å resolution map with no symmetry imposed as described above, and, because of box size limits in cryoSPARC, were downsampled to a 250 pixel box size (~2.2 Å/pix). A new ab initio model was generated from these down sampled particles, and a C1 homogeneous refinement resulted in a 4.6 Å map. 3DVA was then performed using the downsampled particles and mask from the refinement job, three modes, and a filter resolution of 5 Å. The 3DVA display job was run in the simple output mode with 20 frames per clusters. All clusters exhibited the differences in alignment of the C16 dome and the C18 PR about the rotational axis, with the C13 OMC disk held in the same relative position (*Video 4*). The 3DVA display job was then run in the cluster output mode with five clusters. The particles and maps from each cluster were separately subjected to C1 homogeneous refinement. In each case, the resulting 3D maps were at 4.6 Å resolution.

## Model construction and refinement

To construct a model of the OMC, the asymmetric unit of the previously determined structure of the OMC (PDB 6 × 62) was first extracted in Pymol and docked into the map presented here using UCSF Chimera (*Pettersen et al., 2004*). This model was then refined in PHENIX using Phenix.real.space.refine

(*Afonine et al., 2018*). The models were then inspected in Coot, and any subtle differences between the two maps were examined and corrected by hand. Where appropriate, the models were extended in Coot (*Emsley et al., 2010*). Each of the components that were identified in this study (Lpg0823, Lpg2847, and DotF$_1$) were constructed de novo in Coot to generate a model of the entire asymmetric unit. This model was then refined in PHENIX using secondary structure and Ramachandran restraints. The refinement strategy was also optimized by adjusting the nonbonded weighting. A model of the entire OMC was then generated in PHENIX by applying symmetry and further refined as described above. The model was inspected for fit by hand and validated in Phenix using phenix.validation.cryoem. A model of the PR was constructed essentially as described above using as a starting point the previously reported polyalanine models (PDB 6 × 64). The models were adjusted in Coot, symmetrized, and refined in PHENIX essentially as described above to generate a model of the entire PR.

To generate a model of the OMC, the models of the OMC and PR described here (PDBs 7MUC and 7MUE, respectively) were first fit into the map generated without symmetry. The model was then refined in PHENIX and adjusted where necessary in Coot. Models of the DotH linkers were generated by hand in Coot. To model the C-terminal domain of DotH between the OMC and PR, a polyalanine model was first constructed. The DotH C-terminal domain was then aligned to this polyalanine model and refined by hand in Coot. Once completed the asymmetric model was refined in PHENIX.

## Acknowledgements

The work described here was supported by NIH R01AI118932 (MDO), and R21AI6465 (MS and MDO), F32 AI150027-01 (CLD), NIH T32DK007673 (MJS), K99AI154672 (MJS), S10OD020011, S10OD030275, the National Science Foundation DGE1841052 (JR) and the University of Michigan Department of Microbiology and Immunology (Swanson). Some of this work was performed at the Stanford-SLAC Cryo-EM Facilities, supported by Stanford University, SLAC, and the National Institutes of Health S10 Instrumentation Programs. The content is solely the responsibility of the authors and does not necessarily represent the official views of the National Institutes of Health. The molecular graphics and analyses presented here were performed with UCSF Chimera (developed by the Resource for Biocomputing, Visualization, and Informatics at UC-San Francisco, with support from NIH P41-GM103311) and ChimeraX (developed by the Resource for Biocomputing, Visualization, and Informatics at the University of California, San Francisco, with support from National Institutes of Health R01-GM129325 and the Office of Cyber Infrastructure and Computational Biology, National Institute of Allergy and Infectious Diseases). Mass spectrometry experiments were performed by the University of Michigan Proteomics Resource Facility. We thank T Knight for *Legionella* culture advice. We acknowledge the use of the U-M LSI cryo-EM facility, managed by M Su, A Bondy, and L Koepping, and U-M LSI IT support. We thank U-M BSI and LSI for significant support of the cryo-EM facility.

## Additional information

### Funding

| Funder | Grant reference number | Author |
| --- | --- | --- |
| National Institute of Allergy and Infectious Diseases | R01AI118932 | Jeong Min Chung Jacquelyn Roberts Dana Borden Lacy Melanie D Ohi |
| National Institute of Allergy and Infectious Diseases | R21AI6465 | Michele Swanson Melanie D Ohi |
| National Science Foundation | DGE 1841052 | Jacquelyn Roberts |
| National Institute of Allergy and Infectious Diseases | F32 AI150027 | Clarissa L Durie |
| National Institute of Diabetes and Digestive and Kidney Diseases | T32DK007673 | Michael J Sheedlo |

| Funder | Grant reference number | Author |
|---|---|---|
| National Institutes of Health | S10OD030275 | Melanie D Ohi |
| National Institutes of Health | K99AI154672 | Michael J Sheedlo |
| National Institutes of Health | S10OD020011 | Jacquelyn Roberts |
| National Science Foundation | DGE1841052 | Jacquelyn Roberts |

The funders had no role in study design, data collection and interpretation, or the decision to submit the work for publication.

### Author contributions

Michael J Sheedlo, Formal analysis, Investigation, Methodology, Validation, Visualization, Writing - original draft, Writing - review and editing; Clarissa L Durie, Conceptualization, Data curation, Formal analysis, Funding acquisition, Investigation, Methodology, Validation, Visualization, Writing - original draft, Writing - review and editing; Jeong Min Chung, Investigation, Methodology, Visualization; Louise Chang, Investigation, Methodology; Jacquelyn Roberts, Formal analysis, Writing - review and editing; Michele Swanson, Conceptualization, Investigation, Methodology, Resources, Validation, Writing - review and editing; Dana Borden Lacy, Formal analysis, Methodology, Resources, Supervision, Visualization, Writing - original draft, Writing - review and editing; Melanie D Ohi, Conceptualization, Data curation, Formal analysis, Funding acquisition, Investigation, Methodology, Resources, Supervision, Visualization, Writing - original draft, Writing - review and editing

### Author ORCIDs

Michael J Sheedlo ![orcid] http://orcid.org/0000-0002-3185-1727
Clarissa L Durie ![orcid] http://orcid.org/0000-0002-4027-4386
Jeong Min Chung ![orcid] http://orcid.org/0000-0002-4285-8764
Michele Swanson ![orcid] http://orcid.org/0000-0003-2542-0266
Dana Borden Lacy ![orcid] http://orcid.org/0000-0003-2273-8121
Melanie D Ohi ![orcid] http://orcid.org/0000-0003-1750-4793

### Decision letter and Author response

Decision letter https://doi.org/10.7554/eLife.70427.sa1
Author response https://doi.org/10.7554/eLife.70427.sa2

## Additional files

### Supplementary files
• Transparent reporting form

### Data availability

All models and maps have been uploaded to the PDB and the EMDB under accession numbers: PDB 7MUD (EMDB 24005), PDB 7MUE (EMDB 24006), PDB 7MUC (EMDB 24004), PDB 7MUQ (EMDB 24018), PDB 7MUS (EMDB 24020), PDB 7MUV (EMDB 24023), PDB 7MUW (EMDB 24024), PDB 7MUY (EMDB 24026).

The following dataset was generated:

| Author(s) | Year | Dataset title | Dataset URL | Database and Identifier |
|---|---|---|---|---|
| Sheedlo MJ, Durie CL, Swanson M, Lacy DB, Ohi MD | 2021 | Legionella pneumophila Dot/Icm T4SS OMC | https://www.rcsb.org/structure/7MUD | RCSB Protein Data Bank, 7MUD |

*Continued on next page*

*Continued*

| Author(s) | Year | Dataset title | Dataset URL | Database and Identifier |
|---|---|---|---|---|
| Sheedlo MJ, Durie CL, Swanson M, Lacy DB, Ohi MD | 2021 | Legionella pneumophila Dot/Icm T4SS OMC | https://www.ebi.ac.uk/pdbe/entry/emdb/EMD-24005 | Electron Microscopy Data Bank, EMD-24005 |
| Sheedlo MJ, Durie CL, Swanson M, Lacy DB, Ohi MD | 2021 | Legionella pneumophila Dot/Icm T4SS PR | https://www.rcsb.org/structure/7MUE | RCSB Protein Data Bank, 7MUE |
| Sheedlo MJ, Durie CL, Swanson M, Lacy DB, Ohi MD | 2021 | Legionella pneumophila Dot/Icm T4SS PR | https://www.ebi.ac.uk/pdbe/entry/emdb/EMD-24006 | Electron Microscopy Data Bank, EMD-24006 |
| Sheedlo MJ, Durie CL, Swanson M, Lacy DB, Ohi MD | 2021 | Legionella pneumophila Dot/Icm T4SS C1 Reconstruction | https://www.rcsb.org/structure/7MUC | RCSB Protein Data Bank, 7MUC |
| Sheedlo MJ, Durie CL, Swanson M, Lacy DB, Ohi MD | 2021 | Legionella pneumophila Dot/Icm T4SS C1 Reconstruction | https://www.ebi.ac.uk/pdbe/entry/emdb/EMD-24004 | Electron Microscopy Data Bank, EMD-24004 |
| Sheedlo MJ, Durie CL, Swanson M, Lacy DB, Ohi MD | 2021 | Reconstruction of the Legionella pneumophila Dot/Icm T4SS 3DVA Map 1 | https://www.rcsb.org/structure/7MUQ | RCSB Protein Data Bank, 7MUQ |
| Sheedlo MJ, Durie CL, Swanson M, Lacy DB, Ohi MD | 2021 | Reconstruction of the Legionella pneumophila Dot/Icm T4SS 3DVA Map 1 | https://www.ebi.ac.uk/pdbe/entry/emdb/EMD-24018 | Electron Microscopy Data Bank, EMD-24018 |
| Sheedlo MJ, Durie CL, Swanson M, Lacy DB, Ohi MD | 2021 | Reconstruction of the Legionella pneumophila Dot/Icm T4SS 3DVA Map 2 | https://www.rcsb.org/structure/7MUS | RCSB Protein Data Bank, 7MU |
| Sheedlo MJ, Durie CL, Swanson M, Lacy DB, Ohi MD | 2021 | Reconstruction of the Legionella pneumophila Dot/Icm T4SS 3DVA Map 2 | https://www.ebi.ac.uk/pdbe/entry/emdb/EMD-24020 | Electron Microscopy Data Bank, EMD-24020 |
| Sheedlo MJ, Durie CL, Swanson M, Lacy DB, Ohi MD | 2021 | Reconstruction of the Legionella pneumophila Dot/Icm T4SS 3DVA Map 3 | https://www.rcsb.org/structure/7MUV | RCSB Protein Data Bank, 7MUV |
| Sheedlo MJ, Durie CL, Swanson M, Lacy DB, Ohi MD | 2021 | Reconstruction of the Legionella pneumophila Dot/Icm T4SS 3DVA Map 3 | https://www.ebi.ac.uk/pdbe/entry/emdb/EMD-24023 | Electron Microscopy Data Bank, EMD-24023 |
| Sheedlo MJ, Durie CL, Swanson M, Lacy DB, Ohi MD | 2021 | Reconstruction of the Legionella pneumophila Dot/Icm T4SS 3DVA Map 4 | https://www.rcsb.org/structure/7MUW | RCSB Protein Data Bank, 7MUW |
| Sheedlo MJ, Durie CL, Swanson M, Lacy DB, Ohi MD | 2021 | Reconstruction of the Legionella pneumophila Dot/Icm T4SS 3DVA Map 4 | https://www.ebi.ac.uk/pdbe/entry/emdb/EMD-24024 | Electron Microscopy Data Bank, EMD-24024 |
| Sheedlo MJ, Durie CL, Swanson M, Lacy DB, Ohi MD | 2021 | Reconstruction of the Legionella pneumophila Dot/Icm T4SS 3DVA Map 5 | https://www.rcsb.org/structure/7MUY | RCSB Protein Data Bank, 7MUY |
| Sheedlo MJ, Durie CL, Swanson M, Lacy DB, Ohi MD | 2021 | Reconstruction of the Legionella pneumophila Dot/Icm T4SS 3DVA Map 5 | https://www.ebi.ac.uk/pdbe/entry/emdb/EMD-24026 | Electron Microscopy Data Bank, EMD-24026 |

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
