## [Decision Letter]

**Acceptance summary:**

The authors solved the structure of an outer membrane core complex associated with the Legionella Dot/Icm Type IV Secretion System by single particle cryoelectron microscopy. The refined structures identify several novel and important features of the complex. These include three previously unidentified components and a helical extension from the Dis1 subunit that appears to extend vertically and likely anchors the complex to the outer membrane. The paper provides new insights into the Type IV Secretion System, and will thus be of interest to many people working in the area of bacterial pathogenesis.

**Decision letter after peer review:**

Thank you for submitting your article "Cryo-EM reveals new species-specific proteins and symmetry elements in the Legionella pneumophila Dot/Icm T4SS" for consideration by *eLife*. Your article has been reviewed by 2 peer reviewers, and the evaluation has been overseen by a Reviewing Editor and Bavesh Kana as the Senior Editor. The following individuals involved in review of your submission have agreed to reveal their identity: Peter Christie (Reviewer #2).

Essential revisions:

1) The functional analysis to examine the Dot/Icm T4SS-mediated translocation of proteins has not been conducted. This would be required to justify the main conclusion. Otherwise, the main conclusions need to be toned down.

2) The basis of the claim that the intact Dot/Icm T4SS contains 31 copies of DotF (Line 206) and 16 copies of DotG (Line 266) is not clear. Although the C16 dome is mainly composed with DotG in the revealed structure (Figure 5), both DotF and DotG are involved in constituting the C18 PR (Figure 3) as inner membrane-spanning proteins. How can an additional 2 molecules (18-16 = 2) of DotG be supplied in the PR?

3) DotH spans between the C13 OMC and the C18 PR with distinctive copy numbers in three separate planes (Figure 4), and the C-terminal domain of DotH shows remarkable conformational flexibility in the OMC (Figure 4 Supplement 6). On the other hand, the N-terminal domain of DotH seems to stably fit to the C18 PR (Video 1) together with DotG (Figure 3 Supplement 3) and DotF (Figure 3 Supplement 4). This makes one expect that the PR defines the copy number of each component protein. According to the previously defined stoichiometry (Durie et al., 2020), 2:1:1 (DotD: DotC: DotH), can this study determine the possible copy numbers of DotD, DotC, DotH, DotG and DotF per the single Dot/Icm T4SS structure?

4) The functional significance of the components newly identified in this study (Dis2 and Dis3) as well as Dis1 should be elucidated by deleting the genes for applying the effector-translocation assay.

Additional Recommendations:

2. Figure 1. Suppl. 2. Panel A is uninformative and could be deleted.

3. Figure 2. The title implies more than one asymmetric unit but only one is shown in different orientations.

4. Figure 2. Suppl. 2, 3. It would be useful for the reader if the proteins at the left had their names listed immediately above or below.

5. Figure 2 Suppl. 4. The predicted lipidation site of Dis1 is after an unusually long signal sequence, this assignment should be revaluated or better yet confirmed by palmitate labeling, mutagenesis or both.

6. Figure 2, Suppl. 4. Also, the proposal that α-helices constitute transmembrane domains for bacterial outer membranes is not completely heretical, e.g., the VirB10 AP domain, but it is uncommon. Is there a precedent for lipoproteins anchored via N-terminal Cys residues to the OM that then span the OM via adjacent α-helices? The same concern applies for Dis2, which is also proposed to anchor the OMC via a TM α-helix – this is just not a common motif for integral outer membrane proteins, the apparent reason being that there is no Sec system or YidC in the OM to orchestrate TM helix insertion. In this context, it is also not advised to show results of the TM helix prediction algorithm for an OM protein with α-helices.

7. Figure 2, Suppl. 5. Panel e. see above comments.

8. L. 140. The inclusion of Dis1 and Dis2 as anchoring proteins is highly speculative and likely wrong. Mutational analyses should be carried out to evaluate the importance of these a-helices and the C42 residue of Dis1.

9. L. 185. The authors should comment a bit more about the possible significance of DotG segment 791-824 with DotH in the PR. If this contact is real, it suggests that DotG also accommodates the symmetry mismatch between the dome and the PR, do the authors predict these are stable or dynamic interactions?

10. L. 211 and Figure 4. The spatial relationships of DotH in the OMC, intervening space and PR are confusing, principally because the authors are not clear about which domains of DotH comprise these regions. If my interpretation is correct, it should be more clearly stated that there are 18 copies of DotH in the entire structure and that all 18 NTDs comprise the PR. 13 of the associated CTDs extend up to build part of the OMC, while the other 5 extend up only part way into the intervening space. Right?

11. Figure 4. If the interpretation is correct, then why do the authors think that 5 CTDs of DotH floating around between the OMC and PR would confer conformational flexibility to the PR?

12. Lines 234-237 As the complex isolated from L. pneumophila is not in the effector-translocation state (not in the infection condition), I think that the authors may need to tone down the statement about the flexible domain-organization model.

13. Line 62 DotK and Lpg0657 were described as proteins associated with the T4SS core complex not only by their recent report (Durie et al., 2020) but also by Ghosal et al., 2019 (DotK) and Kubori et al., 2014 (in Table S1, DotK and Lpg0657). The original reports should be cited.

14. Line 74 The term of "core" should be more carefully described. Without experimental data showing the functional relevance, the associated proteins which were identified in the structure should not be described as the "core components".

15. Lines 234-237 As the complex isolated from L. pneumophila is not in the effector-translocation state (not in the infection condition), I think that the authors may need to tone down the statement about the flexible domain-organization model.

16. Line 62 DotK and Lpg0657 were described as proteins associated with the T4SS core complex not only by their recent report (Durie et al., 2020) but also by Ghosal et al., 2019 (DotK) and Kubori et al., 2014 (in Table S1, DotK and Lpg0657). The original reports should be cited.

17. Line 74 The term of "core" should be more carefully described. Without experimental data showing the functional relevance, the associated proteins which were identified in the structure should not be described as the "core components".

---

## [Author Response]

Essential revisions:1) The functional analysis to examine the Dot/Icm T4SS-mediated translocation of proteins has not been conducted. This would be required to justify the main conclusion. Otherwise, the main conclusions need to be toned down.

We thank the reviewers for this feedback. We have modified the manuscript to attenuate our conclusions and have made sure to add qualifying statements that explicitly draw the reader’s attention to what must still be considered hypotheses in need of additional experiments. These changes have been made in the “Abstract” and the “Results and Discussion” sections.

2) The basis of the claim that the intact Dot/Icm T4SS contains 31 copies of DotF (Line 206) and 16 copies of DotG (Line 266) is not clear. Although the C16 dome is mainly composed with DotG in the revealed structure (Figure 5), both DotF and DotG are involved in constituting the C18 PR (Figure 3) as inner membrane-spanning proteins. How can an additional 2 molecules (18-16 = 2) of DotG be supplied in the PR?

We apologize for the confusion. We have updated the manuscript to read:

“Based on the structure of DotG modelled within the PR (residues 791-824) and the homology model fit into the dome density (consisting approximately of residues 857-1046), we hypothesize that DotG extends from the PR to the dome, though a physical connection between the two structures is not observed. This would lead to a total of 18 copies of DotG in the intact Dot/Icm T4SS with two copies of DotG not visualized within the dome likely due to structural heterogeneity.”

3) DotH spans between the C13 OMC and the C18 PR with distinctive copy numbers in three separate planes (Figure 4), and the C-terminal domain of DotH shows remarkable conformational flexibility in the OMC (Figure 4 Supplement 6). On the other hand, the N-terminal domain of DotH seems to stably fit to the C18 PR (Video 1) together with DotG (Figure 3 Supplement 3) and DotF (Figure 3 Supplement 4). This makes one expect that the PR defines the copy number of each component protein. According to the previously defined stoichiometry (Durie et al., 2020), 2:1:1 (DotD: DotC: DotH), can this study determine the possible copy numbers of DotD, DotC, DotH, DotG and DotF per the single Dot/Icm T4SS structure?

We thank the reviewers for this suggestion and added the following text to the manuscript directly after the addition made in point 2:

“Taken together, we propose a final stoichiometry of 31:26:18:18:13:13:13:13:13 (DotF:DotD:DotG:DotH:DotC:DotK:Dis1:Dis2:Dis3) for the Dot/Icm T4SS.”

4) The functional significance of the components newly identified in this study (Dis2 and Dis3) as well as Dis1 should be elucidated by deleting the genes for applying the effector-translocation assay.

We thank the reviewers for this feedback and enthusiastically agree that further studies into the functional roles of these proteins are needed. Though these experiments are underway, they require time and resources that are beyond the scope of the current study. As mentioned above, we have modified the manuscript to attenuate our conclusions and have made sure to add qualifying statements that explicitly draw the reader’s attention to what must still be considered hypotheses in need of additional experiments. In addition, we now discuss some additional relevant work by other labs who have published studies on Dis1 and Dis3 function. We have found no studies pertaining to Dis2 function. This is the added text:

** **

“A previous study found that a transposon mutant interrupting the Dis1 gene resulted in a strain that replicates as well as the wild-type strain in liquid culture, but has an intracellular growth defect in both *A. castellanii *and bone marrow-derived murine macrophages, consistent with Dis1 playing an important role in Dot/Icm T4SS function (Goodwin et al., 2016).

Lpg2847 was previously identified as part of a two-gene operon with *lpg2848* or *snrnA*, an RNase secreted by the *L. pneumophila *Type II Secretion System (T2SS). Insertional mutants of both the *srnA* and *dis3* genes were prepared and tested for the ability to secrete the RNase through the T2SS. The *dis3* mutant showed no growth defect in the protozoan host *H. vermiformis*, a well-accepted assay for T2SS function, but has not yet been tested specifically for T4SS assembly or function (Rossier et al., 2009).”

Additional Recommendations:2. Figure 1. Suppl. 2. Panel A is uninformative and could be deleted.

While we thank the reviewer for this feedback, we have elected to keep this panel in the supplemental figure. In our experience, many readers prefer to see a representative micrograph so that they may visually assess the quality of the raw data.

3. Figure 2. The title implies more than one asymmetric unit but only one is shown in different orientations.

We thank the reviewer for catching this typo. It has been corrected to indicate only one asymmetric unit is shown.

4. Figure 2. Suppl. 2, 3. It would be useful for the reader if the proteins at the left had their names listed immediately above or below.

The protein names are now listed directly above the left panels.

5. Figure 2 Suppl. 4. The predicted lipidation site of Dis1 is after an unusually long signal sequence, this assignment should be re-evaluated or better yet confirmed by palmitate labeling, mutagenesis or both.

Since the palmination site in Dis1 has only been predicted (and not actually assigned or observed), the reviewer is correct that naming this site is premature. To address this criticism, we have redone the figure and changed the text so that only lipidation sites that have been previously described in the literature are included in the figure and discussed in the paper.

6. Figure 2, Suppl. 4. Also, the proposal that α-helices constitute transmembrane domains for bacterial outer membranes is not completely heretical, e.g., the VirB10 AP domain, but it is uncommon. Is there a precedent for lipoproteins anchored via N-terminal Cys residues to the OM that then span the OM via adjacent α-helices? The same concern applies for Dis2, which is also proposed to anchor the OMC via a TM α-helix – this is just not a common motif for integral outer membrane proteins, the apparent reason being that there is no Sec system or YidC in the OM to orchestrate TM helix insertion. In this context, it is also not advised to show results of the TM helix prediction algorithm for an OM protein with α-helices.7. Figure 2, Suppl. 5. Panel e. see above comments.8. L. 140. The inclusion of Dis1 and Dis2 as anchoring proteins is highly speculative and likely wrong. Mutational analyses should be carried out to evaluate the importance of these a-helices and the C42 residue of Dis1.

We agree with the reviewer comments (6, 7, and 8) that without additional studies this remains a speculative model. We have now removed these sections from the manuscript since they would require extensive additional studies to test these hypotheses.

9. L. 185. The authors should comment a bit more about the possible significance of DotG segment 791-824 with DotH in the PR. If this contact is real, it suggests that DotG also accommodates the symmetry mismatch between the dome and the PR, do the authors predict these are stable or dynamic interactions?

We agree that the interactions observed between DotG and DotH in the PR are very interesting and may dictate the overall organization of the Dot/Icm T4SS. Unfortunately, we cannot determine the extent of the stability of these interactions from these data. Interestingly, however, similar interactions have been observed within the *X. citri* and *H. pylori* Cag T4SSs which may suggest an important function for this interaction among all T4SSs. We have noted this similarity in the manuscript:

“The interaction that is observed between DotG/DotH in the PR is similar in structure to those previously reported for VirB10/VirB9 and CagX/CagY and likely reflects an important function for retaining this organization within the PR.”

10. L. 211 and Figure 4. The spatial relationships of DotH in the OMC, intervening space and PR are confusing, principally because the authors are not clear about which domains of DotH comprise these regions. If my interpretation is correct, it should be more clearly stated that there are 18 copies of DotH in the entire structure and that all 18 NTDs comprise the PR. 13 of the associated CTDs extend up to build part of the OMC, while the other 5 extend up only part way into the intervening space. Right?

This understanding is correct. We have incorporated a statement outlining the organization/number of the DotH NTDs and CTDs starting at line 234 as follows:

“Thus, there are 18 copies of DotH in the entire structure with all 18 NTDs comprising the PR, 13 of the associated CTDs extending up to build part of the OMC disk, and the other five CTDs extending up only part way into the intervening space.”

11. Figure 4. If the interpretation is correct, then why do the authors think that 5 CTDs of DotH floating around between the OMC and PR would confer conformational flexibility to the PR?

The five DotH CTDs located between the OMC and the PR appear to be dynamic, with changes in orientation occurring between these domains and the rest of the Dot/Icm. We appreciate that this point was not clearly stated in the text. To clarify to the reader which portions of the Dot/Icm appear to be mobile we have added the following text:

“Each of the intervening DotH C-terminal domains between the OMC and PR occurs every two to three asymmetric units, and the degree to which they can be observed varies, indicating that the position of these domains is not static with respect to the PR or the OMC (Figure 4B and Video 1). The flexibility of these five DotH CTDs suggests a mechanism through which the symmetry mismatch observed between the OMC and PR can be accommodated though the utility of the symmetry mismatch cannot be inferred.”

12. Lines 234-237 As the complex isolated from L. pneumophila is not in the effector-translocation state (not in the infection condition), I think that the authors may need to tone down the statement about the flexible domain-organization model.

We have taken the reviewers advice and replaced the text “With this understanding of how the symmetry mismatch is accommodated in the Dot/Icm T4SS, we propose that flexible and/or dynamic connection between regions of PR and OMC, that are not seen in the Cag T4SS will be important for the Dot/Icm T4SS translocating such a uniquely large repertoire of secretion substrates” with:

“Although we are now in a position to describe how the symmetry mismatch is accommodated, its impact on function remains to be determined.”

13. Line 62 DotK and Lpg0657 were described as proteins associated with the T4SS core complex not only by their recent report (Durie et al., 2020) but also by Ghosal et al., 2019 (DotK) and Kubori et al., 2014 (in Table S1, DotK and Lpg0657). The original reports should be cited.

We thank the reviewer for bringing our attention to this unintentional oversite. We have now cited all the appropriate references and did not mean to leave out the important original reports.

14. Line 74 The term of "core" should be more carefully described. Without experimental data showing the functional relevance, the associated proteins which were identified in the structure should not be described as the "core components".

The word “core” has been deleted from this line so that the statement now only refers to “new components” rather than “new core components.”

15. Lines 234-237 As the complex isolated from L. pneumophila is not in the effector-translocation state (not in the infection condition), I think that the authors may need to tone down the statement about the flexible domain-organization model.

This comment has been addressed (point 12 above).

16. Line 62 DotK and Lpg0657 were described as proteins associated with the T4SS core complex not only by their recent report (Durie et al., 2020) but also by Ghosal et al., 2019 (DotK) and Kubori et al., 2014 (in Table S1, DotK and Lpg0657). The original reports should be cited.

This comment has been addressed (point 13 above).

17. Line 74 The term of "core" should be more carefully described. Without experimental data showing the functional relevance, the associated proteins which were identified in the structure should not be described as the "core components".

This comment has been addressed (point 14 above).